# Locating the evidence for children and young people social prescribing: Where to start? A scoping review protocol

Julie Feather[1]*, Shaun Liverpool[2,3], Eve Allen[2,3], Michael Owen[2,3], Nicola Relph[2,3], Lynsey Roocroft[2,3], Tasneem Patel[2,3], Hayley McKenzie[2,3], Ciaran Murphy[2], Michelle Howarth[2,3]

1 Evaluation and Policy Analysis Unit, Edge Hill University, Ormskirk, Lancashire, United Kingdom, 2 Faculty of Health, Social Care and Medicine, Edge Hill University, Ormskirk, Lancashire, United Kingdom, 3 Health Research Institute, Edge Hill University, Ormskirk, Lancashire, United Kingdom

* featherj@edgehill.ac.uk

**Data Availability Statement:** No datasets were generated or analysed during the current study. All relevant data from this study will be made available upon study completion.

## Abstract

It is estimated that disruptions to life caused by the COVID-19 pandemic have led to an increase in the number of children and young people suffering from mental health issues globally. In England one in four children experienced poor mental health in 2022. Social prescribing is gaining traction as a systems-based approach, which builds upon person-centered methods, to refer children and young people with non-clinical mental health issues to appropriate community assets. Recognition of social prescribing benefits for children's mental health is increasing, yet evidence is limited. Inconsistent terminology and variation of terms used to describe social prescribing practices across the literature hinders understanding and assessment of social prescribing's impact on children's mental health. This scoping review thus aims to systematically identify and analyse the various terms, concepts and language used to describe social prescribing with children and young people across the wider health and social care literature base. The scoping review will be undertaken using a six-stage framework which includes: identifying the research question, identifying relevant studies, study selection, charting the data, collating, summarising and reporting the results, and consultation. Electronic databases (MEDLINE, Embase, Cumulative Index to Nursing and Allied Health, PsychInfo, Social Policy Practice, Scopus, Science Direct, Cochrane library and Joanna Briggs), alongside evidence from grey literature, hand search, citation tracking, and use of expert correspondence will be included in the review to ensure published and unpublished literature is captured. Data extraction will be carried out by two reviewers using a predefined form to capture study characteristics, intervention descriptions, outcomes, and key terms used to report social prescribing for children and young people. No formal quality appraisal or risk of bias evaluation will be performed, as this scoping review aims to map and describe the literature. Data will be stored and managed using the Rayaan.ai platform and a critical narrative of the common themes found will be included.

**Funding:** The author(s) received no specific funding for this work.

**Competing interests:** The authors have declared that no competing interests exist.

## Introduction

In the post COVID-19 era, the need to empower resilience in communities and individuals is key to tackling widening health inequalities. Marmot et al. [1] recommended a cross departmental health inequalities strategy which could lay the foundation for a new social contract. The Hewitt Review [2] also advocates a paradigm shift that promotes health and wellbeing through upstream approaches. Recently, the United Kingdom (UK) National Health Service (NHS) Long Term Plan [3] promoted social prescribing to help reduce tackling health inequalities. According to the Kings Fund [4] "*Social prescribing, also sometimes known as community referral, is a means of enabling health [and care] professionals to refer people to a range of local, non-clinical services. The referrals generally, but not exclusively, come from professionals working in primary care settings, for example, GPs or practice nurses*". A global definition, recently developed by Muhl et al. [5 p8] further encapsulates the definition as a concept as being "*a means for trusted individuals in clinical and community settings to identify that a person has non-medical, health-related social needs and to subsequently connect them to non-clinical supports and services within the community by co-producing a social prescription—a non-medical prescription, to improve health and well-being and to strengthen community connections*". While the practice of referring individuals into community-based support has a long history, since 2016, social prescribing has been increasingly used as an integrated approach by health professionals to refer children and young people for non-clinical reasons to a link worker to assess what matters to an individual before onward referral to a community-based asset.

Social prescribing is increasingly promoted by policy makers and commissioners as a strengths-based approach to supporting communities and is particularly relevant post COVID-19, where it is recognised to have impacted on children and young people's mental health [6]. Indeed, one in four children in England now experience poor mental health [7]. A report undertaken by the Children's Commissioner for England [8] showed that one in nine children aged between 5–19 years had a probable mental health disorder in 2017. This figure increased to one in six in 2019/2020 [6] then to one in four in 2022 [7], leading to a 44% increase in the number of children receiving support from the UK National Health Service (NHS) over just a five-year period. The impact of COVID-19 on children and young people's mental health is echoed globally. Therefore, the World Health Organisation [9] has called for action on children and young people's mental health across health systems.

The 'Are we listening' report conducted by the Care Quality Commission [10] in the UK highlights the importance of children's voices in informing mental health support, stating that services are at crisis point. The report highlighted issues with service access, high demand, and long waiting lists, leading children and young people to need more intensive treatments or feel they must be 'suicidal' to get an appointment [10]. Children and young people often feel they need to be desperate to access mental health support [10]. Future in Mind [11], The Five Year Forward View for Mental Health [12] and Transforming Children and Young People's Mental Health Provision: A Green Paper [13], all discuss improving mental health systems for children and young people in the UK. Despite aims to build the mental health workforce, support school mental health teams, and improve service access, daily barriers still impede children and young people from accessing support [14, 15]. Some of these barriers could be alleviated by taking an upstream approach which includes social prescribing.

When considering access to services, General practitioners (GPs) are key in signposting services [16], but face barriers such as time, confidence, lack of resources, lack of providers and long waiting times [14, 16, 17]. The lack of targeted support can lead to children and young people trying to manage or cope in their own way or question the seriousness of their

challenges [15, 16]. Additionally, children and young people feel that choice is crucial in accessing support [14, 18]. However, choice can be limited by parents, carers, or teachers who may be guided by other professionals or policies [14]. Empowering children and young people with choice, autonomy, education and support can help them make positive changes and feel secure in their decisions [19]. Lack of understanding and education on mental health issues for young people and where to access support is a significant barrier [14, 16, 18]. Again, a paradigm shift, to an upstream preventative approach, such as social prescribing could provide children and young people with more control and access to treatments without needing a gatekeeper.

Within the UK, Social Prescribing emerged in the General Practice Forward View [20] as an approach that health professionals could use to support patients with a non-clinical need. The premise was to introduce a system that facilitated a structured referral enabling patients to engage with local assets to improve wellbeing. As part of this vision, social prescribing models were introduced, supporting holistic health by linking social care and other support needs. Since this time, and to help tackle the impact of health and social inequalities, the NHS Long Term Plan [3] expanded the ambitions to further improve prevention and early intervention. A range of social prescribing models have been developed as a result and are increasingly becoming more embedded within the integrated care systems.

There is growing evidence supporting the development and evaluation of social prescribing programmes across the lifespan. Foremost, social prescribing has proven to have a positive impact on health and well-being as well as service users' experience [21]. Similar positive outcomes have been noted for loneliness, quality of life, self-efficacy, and health care utilisation [22]. Among the older adult population, there is a wealth of research highlighting the positive effects of social prescribing on physical and psychosocial outcomes [23], yet the evidence base for children and young people appears to be limited [24].

A recent rapid review undertaken by Hayes et al. [25] reported that there is some emerging evidence indicating benefits of social prescribing for children and young people growing in the past 2 years from no studies to four studies. Whilst this suggests an interest, it also is indicative of the lack of evidence in this area. Variation and lack of consistency in the terms used to describe social prescribing for children and young people across the wider literature could explain the limited evidence available. For example, common terms in this area include 'social prescribing', 'non-medical intervention', 'community referral' [26], 'community asset', 'community-based support' and 'Arts on Prescription' [27]. Additionally, specific interventions or models of social prescribing for children and young people may use different names or descriptors depending on the context of the intervention. For example, arts-based interventions and physical activity programmes delivered to improve children and young peoples' mental health are often based on the principles of social prescribing yet the term is not explicitly used when reporting on such programmes [27, 28]. The terms and language used to describe social prescribing with children and young people are thus diverse and heterogenous making it difficult to capture the scope of breadth of social prescribing practices for children and young people in the wider literature.

To address this gap, we will undertake a scoping review of the literature using Arksey and O'Malley's [29] stepped framework which facilitates a systematic approach to assess the breadth of evidence around a broad subject. By conducting a scoping review of all available evidence that refers to any form of social prescribing, with children and young people, we aim to systematically identify and analyse the various terms, concepts, and language used across different studies and contexts. This comprehensive approach will enable us to capture the breadth of literature on social prescribing for children and young people, shedding light on diverse practices and perspectives within the wider health and social care landscape. This

scoping review will seek to address the gaps in knowledge and terminology used in social prescribing with children and young people, facilitating more nuanced and informed discussions, and guiding future research and practice in social prescribing for children and young people's mental health. The scoping review will be conducted in accordance with the Preferred Reporting Items for Scoping Reviews (PRISMA-ScR) [30].

## Review questions

1. How is social prescribing with children and young people (up to 19 years old) to improve mental health and wellbeing described within the evidence base?

2. What are the key terms used to report social prescribing with children and young people (up to 19 years old) to improve mental health and wellbeing within the evidence base?

## Eligibility criteria

### Population

The approach to the scoping search will be interactive, as advocated by Brettle and Grant [31]. We will use the Population, Concept, Context (PCC) model to structure search terms [32]. The population under study will include any child or young person up to the age of 19 who has been 'socially prescribed' to receive a non-medical/asset-based intervention to improve their mental health or wellbeing.

### Concept

The principal concepts of interest in this review are 'social prescribing' and 'mental health or wellbeing'. The review will use Muhl et al's [5] definition of social prescribing as "*a means for trusted individuals in clinical and community settings to identify that a person has non-medical, health-related social needs and to subsequently connect them to non-clinical supports and services within the community by co-producing a social prescription—a non-medical prescription, to improve health and well-being and to strengthen community connections*".

Conceptional conditions for inclusion will include:

1. There has been a 'referral' from a health, care or teaching professional or equivalent.

2. The individual has been referred for a non-clinical reason.

3. The referral is to support the child/young person's mental health.

4. The range of assets used can vary and will not be standardised.

Evidence sources that explore a concept that does not meet these conditions will be excluded.

### Context

The context for this review involves clinical and community settings where trusted individuals, such as health, care, or teaching professionals, identify non-medical, health-related social needs and refer children and young people to non-clinical supports and services within the community. This encompasses primary care settings, community programmes, and other non-clinical environments aimed at improving the mental health and wellbeing of children and young people. We will include studies where referrals are made from these settings to community programmes, even if the individual is later admitted to an inpatient setting.

**Table 1. Inclusion and exclusion criteria.**

| Inclusion Criteria | Exclusion Criteria |
|---|---|
| Post publication from 2016 when the term social prescribing was used consistently<br>All types of published and unpublished papers that include key terms used to report social prescribing:<br>• Social prescribing<br>• Non-medical intervention<br>• Non pharmaceutical intervention<br>• Community referral<br>• Referral<br>• Non-clinical services<br>Papers that report on 'social prescription' to receive a non-medical/asset-based intervention for children and young people (up to 19 years old) with the aim to improve mental health or wellbeing.<br>Clinical and community settings including primary care, community programmes and other non-clinical environments.<br>Primary and secondary studies including qualitative, quantitative, mixed methods and reviews<br>Papers published in, or able to be translated to English language | Published before 2016<br>Papers that lack inclusion of key terms<br>Papers that focus on medical treatments instead of non-medical or asset-based interventions, those that deal with adults rather than children and young people up to 19 years old, or that do not address mental health or wellbeing.<br>Studies conducted solely in hospitals, emergency departments, inpatient settings, or inpatient mental health settings, as well as studies that do not involve social prescribing in the community context.<br>Protocol papers, editorials, commentaries and conference abstracts.<br>Papers that cannot be translated to English Language<br>Papers where the full text is unavailable |

However, we will exclude studies conducted solely in hospitals, emergency departments, inpatient settings, or inpatient mental health settings, as well as studies that do not involve social prescribing in the community context.

Inclusion and exclusion criteria for the review can be found in Table 1.

## Methods

This protocol has been developed based on Arksey and O'Malley's [29] framework for scoping reviews which will enable selection of evidence from a range of existing sources. Arksey and O'Malley's [29] framework follows six key stages which ensure a robust approach to identifying all relevant literature. The stages are: 1. Identifying the research question. 2. Identifying relevant studies. 3. Study selection. 4. Charting the data. 5. Collating, summarising and reporting the results. 6. Consultation. According to Howarth et al. [33] scoping reviews enable the synthesis of evidence on a broad subject area allowing the development of logic models and guidelines to support decision making.

### Search strategy

The search strategy aims to locate all types of published and unpublished literature. Electronic searches will include databases MEDLINE, Embase, CINAHL, PsychInfo and Social Policy and Practice. Reviewers will also search Scopus, Science Direct, Cochrane and Joanna Briggs Systematic review databases. An additional search of the grey literature will be undertaken, alongside hand searching the reference lists of relevant papers, citation tracking and use of expert correspondence. Furthermore, a search of the following indicative websites will be undertaken: British Association of Social Work, Association of Child Protection Professionals, Social Work, NHS Digital, NHS Spine, and other credible webpages (e.g. Kings Fund, The Health Foundation). Expert consultation will be sought to ensure data capture of relevant non-published and published literature. We will identify and contact key individuals within Public Health England Children, Young People and Families, Youth Social Prescribing Network, Child Outcomes Research Consortium, Headstart National Evaluation Programme, the Social

**Table 2. Example search terms.**

| Population | Concept | Context |
|---|---|---|
| "Child*" OR "Childhood" OR "Adoles*" OR "Young adult*" OR "Young person*" OR "Young people" OR "Youth*" OR "Teen*" | "Social prescri*" OR "non-medical intervention*" OR "non pharmaceutical intervention*" OR "community referral*" OR, "referral*" OR "non-clinical service*" OR "asset-based*" OR "community-based support*" AND "Mental health" OR "wellbeing*" OR "anxiety*" OR "depression*" OR "social isolation" OR "social anxiety" | "Community-based support*" OR "primary care*" OR "community program*" OR "non-clinical service*" OR "education*" |

Prescribing Network, the Social Prescribing Youth Network, the Global Social Prescribing Alliance and the National Academy for Social Prescribing through existing networks and official communication channels. We will ask for their input on relevant literature and feedback on our review. If there is no response, we will reach out to alternative contacts and extend our consultation to additional relevant organisations and networks. To capture the variance in education and referral systems, we will search UK and international literature. The key words that will guide these searches can be found in Table 2.

## Types of evidence sources

A scoping review will enable us to review all types of evidence. This includes primary research studies, systematic and scoping reviews, grey literature including reports, government documents, policy papers and other non-peer reviewed publications and online sources that may provide insights related to the review question. We will only include sources with full text.

## Evidence screening and selection

Data will be stored and managed using the Rayaan.ai platform (Rayyan - AI Powered Tool for Systematic Literature Reviews). The titles and abstracts of all located papers will be screened by two separate reviewers and assessed according to predetermined inclusion criteria (see Table 1). Papers meeting the inclusion criteria will be retrieved in full and comprehensively assessed by two reviewers independently. Any conflicts will be resolved by discussion with the wider team. Papers failing to meet the inclusion criteria will be excluded from further consideration.

## Data extraction

Two reviewers will extract relevant data from included papers using a data extraction tool (Table 3) designed by the research team. Data extracted will include: (1) bibliographic information including authors, publication year, title, journal or source; (2) study population including age of children and young people and sample size; (3) Aim(s) of study; (4) study setting; (5) study design and data collection methods; (6) type of intervention, programme or service offered to children and young people; (7) key findings and key words used to report social prescribing with children and young people. The data extraction tool may be adapted

**Table 3. Data extraction table.**

| Author name, year of publication, title, and source | Study population (age of children and young people, sample size) | Aim(s) of study | Study setting | Study design and data collection methods | Intervention/programme/service used to support children and young people social prescribing in mental health (e.g. tools, mechanisms, outcomes used for referral) | Key findings/key words used to report social prescribing |
|---|---|---|---|---|---|---|

accordingly as data emerges. Disagreements will be resolved through discussion and, if unresolved, reviewers will seek input from a third reviewer. If needed, the primary author of included studies will be contacted to attain missing data and for clarification of study methods and results. Any changes to the extraction process will be documented in the full review write-up.

## Data analysis and presentation

We will analyse and present our findings using both written narratives and tabular summaries. We will use descriptive statistics to quantify the frequency of key terms related to social prescribing. We will use thematic and content analyses to identify common themes and variations in how social prescribing is described across studies. We will provide illustrative quotes to highlight different interpretations of social prescribing. To validate our findings, we will consult with experts in the field as described in the next section of this protocol.

## Consultation

In order to ensure our findings are aligned with the wider literature, consultations will be undertaken with key stakeholders. The consultations are designed primarily to inform and validate findings from the review. However, it may be necessary to obtain input at earlier stages to sensitise the review team to issues that may or may not appear in the literature, and to be guided towards relevant studies. Representatives from a range of organisations, including Streetgames, GP services and Social Prescribing Task Network will be consulted. We also aim to capture the views of parents and carers and young people with lived experience of mental health problems based on discussions from patient and public involvement and engagement (PPIE) sessions that members of the research team have previously undertaken. Where possible, PPIE consultations will be integrated into both early and later stages of the review process. Initially, we will engage with a small group of parents, carers and young people to seek early feedback to help shape the reviews focus and search strategy. PPIE members will be identified through existing networks and prior engagement sessions conducted by the research team. Consultations will be conducted individually or in groups, and notes will be taken solely to inform the review process. Following the completion of the review, additional PPIE consultations will be conducted to validate findings and ensure that they align with the experiences of parents, carers and young people. We aim to conduct five to seven semi-structured consultations, lasting 1 hour and involving five to ten participants. Although formal ethics approvals will not be required, the consultation exercises will be guided by established ethical codes of conduct [34]. Ethical considerations, including informed consent and confidentiality, will guide these consultations, ensuring stakeholder input shapes the review process and conclusions.

## Supporting information

**S1 Checklist. PRISMA-P 2015 checklist.**
(DOCX)

## Author Contributions

**Conceptualization:** Shaun Liverpool, Michael Owen, Lynsey Roocroft, Hayley McKenzie, Michelle Howarth.

**Methodology:** Julie Feather, Shaun Liverpool, Eve Allen, Michael Owen, Nicola Relph, Lynsey Roocroft, Tasneem Patel, Hayley McKenzie, Ciaran Murphy, Michelle Howarth.

**Writing – original draft:** Julie Feather, Shaun Liverpool, Eve Allen, Michael Owen, Nicola Relph, Lynsey Roocroft, Tasneem Patel, Hayley McKenzie, Ciaran Murphy, Michelle Howarth.

**Writing – review & editing:** Julie Feather, Shaun Liverpool, Eve Allen, Michael Owen, Nicola Relph, Lynsey Roocroft, Tasneem Patel, Hayley McKenzie, Ciaran Murphy, Michelle Howarth.

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
