## [Decision Letter · Decision Letter 0]

9 Jul 2024

PONE-D-24-11234Locating the evidence for children and young people social prescribing: Where to start? A scoping review protocolPLOS ONE

Dear Dr. Feather,

Thank you for submitting your manuscript to PLOS ONE. After careful consideration, we feel that it has merit but does not fully meet PLOS ONE’s publication criteria as it currently stands. Therefore, we invite you to submit a revised version of the manuscript that addresses the points raised during the review process.

** **

This paper outlines a protocol for a scoping review on social prescribing for mental health conditions among young people. It is well written and provides an important contribution to the literature. There are some methodological details missing from this paper that should be addressed by the authors. There are also grammatical and punctuation errors that require the authors' attention.

We look forward to receiving your revised manuscript.

Kind regards,

Apurva kumar Pandya, PhD

Academic Editor

PLOS ONE

Journal Requirements:

**Additional Editor Comments:**

This paper outlines a protocol for a scoping review on social prescribing for mental health conditions among young people. It is well written and provides an important contribution to the literature. There are some methodological details missing from this paper which should be addressed by authors. There are also grammatical and punctuation errors that require authors' attention.

Reviewers' comments:

Reviewer's Responses to Questions

**Comments to the Author**

1. Does the manuscript provide a valid rationale for the proposed study, with clearly identified and justified research questions?

Reviewer #1: Yes

Reviewer #2: Yes

2. Is the protocol technically sound and planned in a manner that will lead to a meaningful outcome and allow testing the stated hypotheses?

Reviewer #1: Partly

Reviewer #2: Yes

3. Is the methodology feasible and described in sufficient detail to allow the work to be replicable?

Reviewer #1: Yes

Reviewer #2: Yes

4. Have the authors described where all data underlying the findings will be made available when the study is complete?

Reviewer #1: No

Reviewer #2: Yes

5. Is the manuscript presented in an intelligible fashion and written in standard English?

Reviewer #1: Yes

Reviewer #2: Yes

6. Review Comments to the Author

You may also provide optional suggestions and comments to authors that they might find helpful in planning their study.

Reviewer #1: Overview: This paper outlines a protocol for a scoping review on social prescribing for mental health conditions among young people. It is well written and provides an important contribution to the literature. There are some methodological details missing from this paper which should be addressed by authors.

Abstract and Introduction:

- Line 25: can remove the (‘) from ‘systems-based approach’

- Line 32: Please include in the abstract the type of review that will be conducted (scoping?)

- Line 39-40: I’m not sure what is meant by ‘data extraction will be carried out by two reviewers to identify the scope and depth of MeSH used in the literature…’; MeSH headings are typically a part of the search strategy and not something you would look to during data extraction so it is unclear what is meant by this sentence. Please clarify.

- Line 42-43: Please include an additional line about data synthesis and risk of bias. How will this step be completed? Will you conduct a risk of bias/quality appraisal assessment?

- The introduction section could be shortened to ensure conciseness; I think the 2nd last paragraph could be removed as it seems a bit repetitive

- Line 72-75: please spell out numbers under 10 (or use % if it’s easier to indicate what is meant)

Methods

- Line 173: could you provide a reference for the 0-25 age range? Other criteria for youth in the literature include 0-18- or 0–19-year-olds, so could authors clarify whether age 25 still counts as youth/young person?

- Line 171-174: could authors add more detail about who is and is not included in this study (do they need to have a diagnosed mental health condition? What about things like eating disorders? What about other neurological or intellectual impairments? What if people have multiple conditions? Would you include any gender, age, ethnicity, any country, etc. What about pregnant individuals? People using alcohol or drugs?)

- Line 191-194: This section should only focus on context (ie. Clinical setting/environment), so any details about participants and social prescribing should be removed from this section. Instead, more detail about what contexts are and are not considered for inclusion is needed (e.g., what about hospitals? Emergency departments? Inpatient settings? In-patient mental health settings? Will you consider studies where the individuals are admitted to an inpatient setting but they have been referred to a community program or will you only consider studies that take place in the setting where they were referred to via social prescribing? What about primary care, community programs, etc.). Please add more detail about what settings are in versus out.

- Table 1 should include all the inclusion/exclusion criteria (for PCC, not just for the types of studies)

- Will you consider studies that are not in English but can be translated to English?

- What about study design? Please outline what types of studies you will vs won’t include (reviews? Protcols? Grey literature?)

- Line 219: How will you complete the expert consultation? Do you know individuals in these organizations who are willing to provide input or are you going to reach out and see who responds? What exactly will their role be? What if they don’t respond or cooperate?

- Table 2: Do you have a librarian to help with the search strategy? Based on table 2, there are some notable limitations with the proposed search. For example, authors should use truncation to capture any related/similar terms, such as adoles* to capture adolescent, adolescence, and adolescents. Depending on the database, you should also use MeSH terms to capture certain topics. Please review the search strategy with a librarian and include an example of a search that will be conducted in one of your databases, like CINAHL, and include this in your table 2

- Table 3: some other details to include: study design, age of participants, study setting

- Table 3: Can you clarify what is meant by MeSH terms in this table? What if studies don’t use MeSH terms (this is data-base dependent), or do you mean ‘key words’? I’m not sure this information will be available for every paper so I’m not sure what value it adds. Rather, you might want to consider a deeper analysis into the way that social prescribing is described in studies if you want to understand how it is reported. Is there a social prescribing framework or definition you could use to inform this?

- Line 261: should say ‘synthesize’ not ‘synthesis’

- Line 261-264: See previous comments about MeSH terms

- Line 261-264: more detail about the synthesis approach is needed; Will authors report findings in writing, in tables, both? Are there other ways to describe how social prescribing is outlined across papers? How will you determine whether studies truly used social prescribing or not?

- Consultation: I like the inclusion of patients/families here! Can you include detail about when this step will take place? Will stakeholders be identified at the outset or will consultations only happen after the review is complete and findings shared with these groups?

Reviewer #2: This paper is a protocol of a scoping review on social prescribing for the mental health of young people and children aged up to 25 years old. The manuscript is well written and the protocol for the scoping review is thorough and technically sound. The rationale for the study is clear and valid. The methodology is reported in detail. The additional step of consultation is excellent and adds to the validity of the study.

Some aspects of the protocol that can be made better:

1. While the methods sections is extremely well written, succinct and clear, I I feel that the Introduction section of the paper could be improved, especially in making it succinct and clearer. For example, on one hand, paragraph 3 and paragraph 4 (paragraphs about young people’s mental health) can be considerably shortened. On the other hand, the paragraphs specifically talking about social prescribing can be improved – for example, for readers unfamiliar with the concept of social prescribing, a brief overview of history of the concept across different countries would be important. Currently, the manuscript gives the impression that social prescribing emerged only around 2016. It would be important to clarify that while the term is relatively new the practice is much older.

2. Specific details on the planned consultation would be useful including how many consultations, of what duration, with how many people, how will they be structured, how will they inform the protocol, how will they inform the review. While it makes sense that an IRB approval may not be required for the consultation, it is important to outline the ethical considerations for these consultations.

7. PLOS authors have the option to publish the peer review history of their article (what does this mean?). If published, this will include your full peer review and any attached files.

Reviewer #1: No

Reviewer #2: No

---

## [Author Response · Author response to Decision Letter 0]

16 Aug 2024

Dear Reviewers,

Thank you for your reply regarding our manuscript PONE-D-24-11234 entitled “Locating the evidence for children and young people social prescribing: Where to start? A scoping review protocol”.

We would like to thank both reviewers for your valuable feedback and observations which have been used to improve our manuscript. We have revised and modified the manuscript in line with your comments.

The table below provides a point-by-point response to the reviewers’ comments. Revisions to the manuscript have been highlighted using the track changes mode in MS Word.

Reviewers’ comments Authors’ response

Reviewer 1 

Abstract:

- Line 25: can remove the (‘) from ‘systems-based approach’

- Line 32: Please include in the abstract the type of review that will be conducted (scoping?)

- Line 39-40: I’m not sure what is meant by ‘data extraction will be carried out by two reviewers to identify the scope and depth of MeSH used in the literature…’; MeSH headings are typically a part of the search strategy and not something you would look to during data extraction so it is unclear what is meant by this sentence. Please clarify.

- Line 42-43: Please include an additional line about data synthesis and risk of bias. How will this step be completed? Will you conduct a risk of bias/quality appraisal assessment? Many thanks for your helpful suggestions on how to improve our abstract. 

Response: 

We have removed (‘) from ‘systems-based approach’. 

We have also added ‘scoping review’ to identify the type of review we will be conducting. 

Line 39-40 has been amended to: Data extraction will be carried out by two reviewers using a predefined form to capture study characteristics, intervention descriptions, outcomes, and terminology related to social prescribing for children and young people.

Line 42-43: We have added the following sentence - No formal quality appraisal or risk of bias evaluation will be performed, as this scoping review aims to map and describe the literature. This is in line with methodological guidance on the conduct of scoping reviews (see Arksey and O’Malley, 2005 and JBI manual for evidence synthesis, 2024, https://jbi-global-wiki.refined.site/space/MANUAL/355862497/10.+Scoping+reviews). 

Introduction:

- The introduction section could be shortened to ensure conciseness; I think the 2nd last paragraph could be removed as it seems a bit repetitive

- Line 72-75: please spell out numbers under 10 (or use % if it’s easier to indicate what is meant) 

Response:

Many thanks for your helpful suggestions. 

We have shortened the introduction section by removing the 2nd to last paragraph. 

We have spelt numbers under 10 out throughout our manuscript. 

Methods:

- Line 173: could you provide a reference for the 0-25 age range? Other criteria for youth in the literature include 0-18- or 0–19-year-olds, so could authors clarify whether age 25 still counts as youth/young person?

- Line 171-174: could authors add more detail about who is and is not included in this study (do they need to have a diagnosed mental health condition? What about things like eating disorders? What about other neurological or intellectual impairments? What if people have multiple conditions? Would you include any gender, age, ethnicity, any country, etc. What about pregnant individuals? People using alcohol or drugs?)

- Line 191-194: This section should only focus on context (ie. Clinical setting/environment), so any details about participants and social prescribing should be removed from this section. Instead, more detail about what contexts are and are not considered for inclusion is needed (e.g., what about hospitals? Emergency departments? Inpatient settings? In-patient mental health settings? Will you consider studies where the individuals are admitted to an inpatient setting but they have been referred to a community program or will you only consider studies that take place in the setting where they were referred to via social prescribing? What about primary care, community programs, etc.). Please add more detail about what settings are in versus out.

- Table 1 should include all the inclusion/exclusion criteria (for PCC, not just for the types of studies)

- Will you consider studies that are not in English but can be translated to English?

- What about study design? Please outline what types of studies you will vs won’t include (reviews? Protocols? Grey literature?)

- Line 219: How will you complete the expert consultation? Do you know individuals in these organizations who are willing to provide input or are you going to reach out and see who responds? What exactly will their role be? What if they don’t respond or cooperate?

- Table 2: Do you have a librarian to help with the search strategy? Based on table 2, there are some notable limitations with the proposed search. For example, authors should use truncation to capture any related/similar terms, such as adoles* to capture adolescent, adolescence, and adolescents. Depending on the database, you should also use MeSH terms to capture certain topics. Please review the search strategy with a librarian and include an example of a search that will be conducted in one of your databases, like CINAHL, and include this in your table 2

- Table 3: some other details to include: study design, age of participants, study setting

- Table 3: Can you clarify what is meant by MeSH terms in this table? What if studies don’t use MeSH terms (this is data-base dependent), or do you mean ‘key words’? I’m not sure this information will be available for every paper so I’m not sure what value it adds. Rather, you might want to consider a deeper analysis into the way that social prescribing is described in studies if you want to understand how it is reported. Is there a social prescribing framework or definition you could use to inform this?

- Line 261: should say ‘synthesize’ not ‘synthesis’

- Line 261-264: See previous comments about MeSH terms

- Line 261-264: more detail about the synthesis approach is needed; Will authors report findings in writing, in tables, both? Are there other ways to describe how social prescribing is outlined across papers? How will you determine whether studies truly used social prescribing or not?

- Consultation: I like the inclusion of patients/families here! Can you include detail about when this step will take place? Will stakeholders be identified at the outset or will consultations only happen after the review is complete and findings shared with these groups?

Response:

Thank you for your very helpful comments on our methods section. We have made the following changes as recommended by reviewers to strengthen our methods section:

Line 173 – Thank you for raising this important point. We have changed our age criteria to 0-19 which is more consistent with the age range for CYP reported in the social prescribing literature. 

Line 171-174 (Population) – We have changed the wording slightly to reflect that ANY child or young person (up to 19 years) will be included in the review on the condition that they have been ‘socially prescribed’ to receive a non-medical/asset-based intervention. This is regardless of diagnosed mental health conditions, specific conditions like eating disorders, neurological or intellectual impairments, multiple conditions, gender, age, ethnicity, country, pregnancy status, or substance use.

Line 191-194 (Context) – We have amended this paragraph and removed information relating to participants and social prescribing. We have given details of which settings/environments will be included and which will be excluded.

Table 1 has now been updated to ensure all PCC inclusion and exclusion criteria are included. 

Yes. We have changed the inclusion and exclusion criteria relating to this in table 1. We will include papers published in, or able to be translated to English Language. 

We have added information on the types of study designs that will be included and excluded in Table 1: Inclusion and exclusion criteria. We will include empirical studies only including qualitative, quantitative or mixed methods design. We will exclude protocol papers, reviews, editorials, commentaries and conference abstracts.

We have amended the section on expert consultation to include details on how we will contact experts, what their role will be and what our contingency plan is if they do not respond. 

Thank you for your valuable feedback. While we are currently unable to work with a librarian, we have taken your suggestions into account and made updates to our search strategy to enhance its effectiveness. Table 2 has been updated. We have re-structured our search terms in line with the Population, Concept, Context (PCC) model. We have incorporated truncation and related terms into our search strategy. We will use appropriate MeSH terms where applicable to ensure comprehensive coverage of topics. For example, in databases like CINAHL we will include MeSH terms relevant to social prescribing and non-clinical interventions. We believe these adjustments address the reviewer’s concerns and will improve the robustness of our search strategy.

We have updated table 3 to include study design, study population and study setting.

We have removed reference to MeSH terms. We have amended the final column on findings to also include information on the concepts used to describe social prescribing for children and young people in the literature. We will use Muhl’s definition of social prescribing as described on page 9 to inform this. 

The word synthesize has now been used. 

We take on board your feedback that not all studies will use MeSH terms or that these might not be uniformly available. Instead of relying on MeSH terms we will instead identify and use key words and phrases that researchers use within the literature to report or describe social prescribing with children and young people. We will use Muhl’s definition of social prescribing as described on page 9 to guide us. We have replaced reference to MeSH terms with key terms throughout our manuscript. 

We have amended the section on data analysis and presentation taking into account reviewer comments. We have described in more detail how data will be analysed and how our findings will be presented. 

We have updated the section on consultations to provide information on how PPIE members will be identified and at what stages we will undertake PPIE consultations. 

Reviewer 2 

Introduction: 

While the methods sections is extremely well written, succinct and clear, I feel that the Introduction section of the paper could be improved, especially in making it succinct and clearer. For example, on one hand, paragraph 3 and paragraph 4 (paragraphs about young people’s mental health) can be considerably shortened. On the other hand, the paragraphs specifically talking about social prescribing can be improved – for example, for readers unfamiliar with the concept of social prescribing, a brief overview of history of the concept across different countries would be important. Currently, the manuscript gives the impression that social prescribing emerged only around 2016. It would be important to clarify that while the term is relatively new the practice is much older. 

Response:

Thank you for your valuable comments. We have reviewed our introduction and tried where possible to shorten this, removing sentences and words that are not required. Paragraphs 3 and 4 have been shortened considerably.

We have further added a short paragraph to provide more context for readers on the history of social prescribing (see lines 129-137). 

We have changed the wording of the sentence which states that social prescribing only emerged around 2016 to instead highlight that the practice of referring individuals into community-based support has a long history. 

Consultations:

Specific details on the planned consultation would be useful including how many consultations, of what duration, with how many people, how will they be structured, how will they inform the protocol, how will they inform the review. While it makes sense that an IRB approval may not be required for the consultation, it is important to outline the ethical considerations for these consultations. 

Response:

Thank you for your very helpful comments. As stated above we have added more detail to the consultations section to include more information on how many consultations will be completed, for how long, how many participants and how these will be used to inform the review. We have also added a sentence which outlines the ethical considerations. 

I look forward to hearing from you regarding our submission. I am happy to respond to any further questions and comments that you may have.

Yours Sincerely,

Author

---

## [Decision Letter · Decision Letter 1]

5 Sep 2024

Locating the evidence for children and young people social prescribing: Where to start? A scoping review protocol

PONE-D-24-11234R1

Dear Dr. Feather,

We’re pleased to inform you that your manuscript has been judged scientifically suitable for publication and will be formally accepted for publication once it meets all outstanding technical requirements.

Kind regards,

Apurva kumar Pandya, PhD

Academic Editor

PLOS ONE

Reviewers' comments:

Reviewer's Responses to Questions

**Comments to the Author**

1. Does the manuscript provide a valid rationale for the proposed study, with clearly identified and justified research questions?

Reviewer #2: Yes

Reviewer #3: Yes

Reviewer #4: Yes

2. Is the protocol technically sound and planned in a manner that will lead to a meaningful outcome and allow testing the stated hypotheses?

Reviewer #2: Yes

Reviewer #3: Yes

Reviewer #4: Yes

3. Is the methodology feasible and described in sufficient detail to allow the work to be replicable?

Reviewer #2: Yes

Reviewer #3: Yes

Reviewer #4: Yes

4. Have the authors described where all data underlying the findings will be made available when the study is complete?

Reviewer #2: No

Reviewer #3: No

Reviewer #4: Yes

5. Is the manuscript presented in an intelligible fashion and written in standard English?

Reviewer #2: Yes

Reviewer #3: Yes

Reviewer #4: Yes

6. Review Comments to the Author

You may also provide optional suggestions and comments to authors that they might find helpful in planning their study.

Reviewer #2: All earlier comments have been addressed to reasonable satisfaction and the paper is recommended for publication.

Reviewer #3: 1) Abstract is well-written.

2) Introduction is well written, but there is a scope to make it more concise.

3) Methodology is well structured addressing all the comments provided by the reviewers initially.

4) Data extraction section is well-written.

The previous reviewers' comments have been addressed and incorporated correctly.

Reviewer #4: Introduction and Rationale:

The introduction emphasizes the increasing acknowledgment of social prescribing as a valuable approach for addressing mental health issues in children and young people. This section is well-crafted and effectively clarifies the importance of social prescribing as a tool in the mental health landscape, particularly for younger populations. By connecting these points, the introduction lays a strong foundation for understanding the relevance and necessity of integrating social prescribing into mental health care strategies for children and adolescents.

Research Gaps and Justification:

The rationale for utilizing a scoping review methodology is well-founded and thoroughly explained. This approach is justified by the need to map out the existing research landscape, identify gaps, and provide a comprehensive overview of the current knowledge on the topic.

Methodology:

The method is clearly articulated, with each step described in a detailed and understandable manner. The clarity and precision with which the methodology is presented contribute to the overall robustness and credibility of the review, making it a valuable tool for systematically mapping the existing research landscape. The adoption of the six-stage framework for the scoping review is both highly appropriate and skillfully executed. This well-established framework guides the review process through each critical phase, from defining the research question to collating and summarizing the data, ensuring a comprehensive exploration of the topic.

Overall Structure:

The paper is thoughtfully organized and highly readable, with a clear and logical flow that guides the reader through the content seamlessly. Each section is well-defined and contributes meaningfully to the overall narrative, ensuring that the paper is easy to follow. The structure effectively supports the presentation of the research, making complex information accessible and engaging. The coherence and clarity of the organization enhance the reader's understanding, allowing the key points and arguments to be conveyed with impact.

7. PLOS authors have the option to publish the peer review history of their article (what does this mean?). If published, this will include your full peer review and any attached files.

Reviewer #2: **Yes: **Harikeerthan Raghuram

Reviewer #3: No

Reviewer #4: No

---

## [Editor Report · Acceptance letter]

10 Sep 2024

PONE-D-24-11234R1 

PLOS ONE

Dear Dr. Feather, 

I'm pleased to inform you that your manuscript has been deemed suitable for publication in PLOS ONE. Congratulations! Your manuscript is now being handed over to our production team.

Kind regards, 

on behalf of

Dr. Apurva kumar Pandya 

Academic Editor

PLOS ONE